# TOWARDS UNSUPERVISED SPEECH RECOGNITION AT THE SYLLABLE-LEVEL

## ABSTRACT

Training speech recognizers with unpaired speech and text – known as unsupervised speech recognition (UASR) – is a crucial step toward extending ASR to low-resource languages in the long-tail distribution and enabling multimodal learning from non-parallel data. However, existing approaches based on phones often rely on costly resources such as grapheme-to-phoneme converters (G2Ps) and struggle to generalize to languages with ambiguous phoneme boundaries due to training instability. In this paper, we address both challenges by introducing a syllable-level UASR framework based on masked language modeling, which avoids the need for G2P and the instability of GAN-based methods. Our approach achieves up to a 40% relative reduction in character error rate (CER) on LibriSpeech and generalizes effectively to Mandarin, a language that has remained particularly difficult for prior methods. Code will be released upon acceptance.

## 1 INTRODUCTION

Recent advances in self-supervised learning (Baevski et al., 2020; Hsu et al., 2021a; Chen et al., 2022; Chung et al., 2021; Chen et al., 2024; Mohamed et al., 2022) and spoken language modeling (Arora et al., 2025; Chu et al., 2024; Défossez et al., 2024) have enabled voice assistants with increasingly human-like listening and speaking abilities. However, they are far from *language-universal*: most systems support only a handful of languages in the world (Chen et al., 2024; Conneau et al., 2021), due to a lack of large-scale paired speech and text training corpora.

A promising step toward language-universal assistants is to build speech recognizers from *unpaired* speech and text, or *unsupervised speech recognition* (UASR) (Glass, 2012; Baevski et al., 2021; Liu et al., 2023; Tseng et al., 2024). UASR is a fundamental challenge: success would enable downstream tasks such as speech synthesis (Ni et al., 2022; Liu et al., 2022), translation (Wang et al., 2023a), and understanding (Shi et al., 2023), paving the way for general-purpose voice assistants. It is also a central case of *unpaired multimodal learning* (Artetxe et al., 2018b;a; 2019; Lample et al., 2018a;b; Ma et al., 2019; Hoshen & Wolf, 2018), where modalities need to be aligned without parallel data. In the absence of sentence-level alignment, the model must infer higher-level linguistic units–phones, syllables, and words–from raw speech waveforms in conjunction with global text statistics. Thus, UASR provides broader insights into representation learning and multimodal alignment without supervision.

The best existing UASR systems (Baevski et al., 2021; Liu et al., 2023; Tseng et al., 2024) operate at the *phoneme-level*. To this end, they need to convert raw text units, or *graphemes*, to phonemes, or minimal sound units that encode meaning, using a grapheme-to-phoneme convertor (G2P). However, training G2Ps requires resources such as pronunciation dictionaries, which can be time-consuming and labor-intensive to create. Without a G2P, such systems suffer from significant performance degradation due to misalignment between speech and raw text in many languages Liu et al. (2023); Ni et al. (2022). Even when pronunciation dictionaries are available, the system may fail to detect clear phone-level boundaries due to strong co-articulation effects for languages such as Mandarin.

An alternative approach to phone-based models is to build a word-level UASR system, which can be achieved without a G2P. Yet a major concern is the coverage of the system on *rare words*, which can effectively be infinite in vocabulary size, making it significantly harder to acquire than phones. Furthermore, detecting word boundaries requires capturing long-range contextual dependencies in speech, which risk destabilizing segmentation mechanisms effective for phone-level UASR Wang et al. (2023c).

In this work, we propose a third alternative – to build UASR at the *syllable-level*, which can be justified on three grounds. First, unlike words, the number of distinct syllables for a language is finite, which reduces the long-tail token distribution issue and allows for better generalization to unseen words; second, many languages exhibit the best alignment between speech and text at the syllable level instead of the phone or word level. For instance, Mandarin uses characters, which have strong correspondences to spoken syllables; therefore, we expect that a syllable-level UASR system could be more appropriate for UASR than a phoneme-based system, which we also found to be the case empirically.

Last but not least, recent advancement in syllable boundary detection and unit discovery (Baade et al., 2025; Cho et al., 2025) has made it possible to segment raw speech into syllable-like units without any textual supervision, and is often more reliable than unsupervised segmentation methods at the word-level, while capturing most of the word-level semantics Peng & Harwath (2022); Fuchs & Hoshen (2023).

In this paper, we make the following contributions.

1. This paper introduces **SylCipher**, to our knowledge, the first syllable-based UASR system. SylCipher jointly predicts syllable boundaries and embedding tokens from raw speech using a unified self-supervised objective. The learning mechanism avoids adversarial training, making it more stable and less sensitive to hyperparameters.

2. We provide an information-theoretic analysis on the proposed UASR system proving that our training objective achieves perfect distribution matching and zero-error UASR under regularity conditions.

3. We conduct extensive experiments across domains and languages. On LibriSpeech, SylCipher achieves up to 40% relative character error rate (CER) reduction over prior G2P-free UASR methods. On SpokenCOCO, improvements are even larger, demonstrating robustness across domains. On Mandarin, SylCipher achieves 12.2% phone error rate (PER), outperforming GAN-based UASR methods that fail to even converge.

4. We perform careful ablation and error analysis, examining the effects of token vocabulary size, syllabifier choice, and segmentation mechanisms.

**Paper organizations.** Section 2 reviews related work. Section 3 formalizes the syllable-level UASR problem. Section 4 presents SylCipher and its theoretical guarantees. Section 5 reports experiments and ablations. Section 6 concludes with limitations and future work.

## 2   RELATED WORK

**Unsupervised speech recognition**   Early work on UASR assumed the existence of a reliable G2P and formulate the problem as an adversarial game, where a conditional generator predicts a phoneme sequence given a speech waveform, and a discriminator tries to tell real phonemized text apart from the generator's output (Wang et al., 2023b). Within this framework, several works have explored different segmentation mechanisms such as fixed unsupervised phoneme segmentation (Liu et al., 2018), iterative forced alignment (Chen et al., 2019), de-duplication Baevski et al. (2021) and reinforcement learning (Tseng et al., 2024). (Wang et al., 2023c) proposed an alternative formulation of UASR as an explicit distribution matching problem, by matching the lower-order $N$-gram distributions of the generated and real texts. Later work further extended this reformulation to use G2P-free tokens such as words (Wang et al., 2023c; Ni et al., 2025), more expressive architecture such as transformers (Ni et al., 2025) and more general text distribution such as masked prediction probabilities (Ni et al., 2025). We adapt this word-level UASR approach to the syllable-level, and supplement it with a simplified training flow, a more stable differentiable boundary detector and additional learning objectives.

**Syllable-level self-supervised learning**   Early work on syllable-level modeling of speech used signal processing techniques (Zhang & Glass, 2009). To discover higher-level structure and more efficient self-supervised learning (SSL) representations from raw speech, (Peng et al., 2023; Cho et al., 2024) proposed to induce syllabic structure from existing SSL models such as HuBERT (Hsu et al., 2021a). To this end, (Peng et al., 2023) probed the self-attention layers of VG-HuBERT (Peng & Harwath, 2022), a visually grounded SSL model to detect syllable-like feature clusters and further refine such clusters using a min-cut algorithm Shi & Malik (1997). To sidestep the need for visual data, (Cho et al., 2024) employed a speech-only SSL model trained with utterance-level self-distillation from a HuBERT teacher. Due to the indirect manner of such approaches by which syllabic structures are derived, they are often noisy and unreliable. To cope with this issue, recent works (Baade et al., 2025; Cho et al., 2025) proposed a more direct and targeted approach by performing self-distillation at the syllable-level, which significantly improved syllable boundary detection and unit discovery performance by encouraging sharper contrast between within and between-syllable feature frames.

## 3   SYLLABLE-LEVEL UNSUPERVISED SPEECH RECOGNITION

In this section, we formulate syllable-level UASR as follows. Let $X = [X_1, \cdots, X_T] \in \mathcal{X}^T$ be a padded sequence of speech feature vectors and let $Y = [Y_1, \cdots, Y_L] \in \mathcal{Y}^L$ a padded sequence of text tokens in the same language. Since a tokenized speech utterance typically uses more tokens than the text transcription

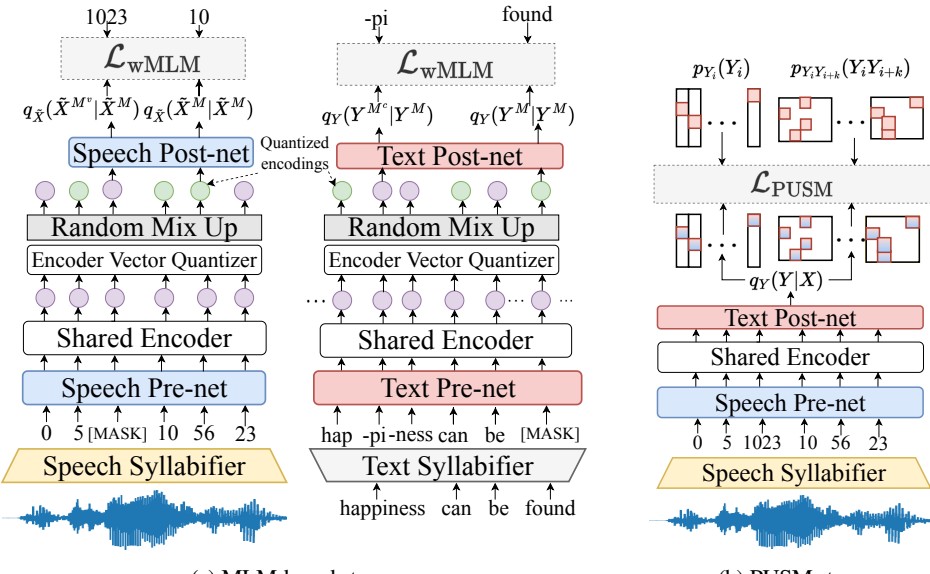

(a) MLM-based stages       (b) PUSM stage

Figure 1: **Overall architecture of SylCipher**. Gray boxes are fixed during training. (a) MLM-based stages: learn a compressed joint semantic space with a shared encoder and random mix-up. (b) PUSM stage: align speech and text spaces by matching lower-order marginals of their distributions.

of the same utterance, we assume the $T \geq L$. Assume $X$ and $Y$ come from two *unpaired* datasets and are therefore *statistically independent*. Further, suppose they are *matched*, i.e., there exists an ASR function $y^* : \mathcal{X} \mapsto \mathcal{Y}$ such that the distributions of $X$ and $Y$, $p_X$ and $p_Y$ satisfy

$$p_Y(y) = \int_{x \in \mathcal{X} : y^*(x) = y} p_X(x)\mathrm{d}x, \forall y \in \mathcal{Y}^L, \tag{1}$$

The goal of UASR is to recover $y^*$ given only unpaired $X$ and $Y$. The syllable-level case is a special setting where the tokens of $Y$ are syllables. The learning problem resembles *decipherment*, where one decodes a message in an unknown script without a lexicon or grammar. In practice, $X$ is formed from frame-level SSL features Hsu et al. (2021a), and $p_X$ and $p_Y$ are only approximately matched due to finite-sample noise and domain mismatch. As shown in (Wang et al., 2023b), UASR is ill-posed in general, but the mapping $y^*$ in equation 1 becomes identifiable if syllable boundaries are known and the language satisfies mild conditions.

## 4 SylCipher: Syllable-level UASR via information-constrained masked language modeling

In this section, we describe **SylCipher**, our proposed model for syllable-level UASR. We first present its architecture and training objective, then justify the design theoretically, and finally introduce several practical modifications for training and inference.

### 4.1 Training: UASR via information compression

As shown in Figure 1, SylCipher is an encoder-only language model with a *shared encoder* for speech and text modalities. To project both modalities into a joint embedding space, we use two uni-modal *pre-nets*: $e_{\tilde{X}} : \mathcal{X}^T \mapsto \mathbb{R}^{L \times d}$ for speech and $e_Y : \mathcal{Y}^L \mapsto \mathbb{R}^{L \times d}$ for text, each implemented as a linear embedding layer. Before the speech pre-net, a *speech syllabifier* converts the frame-level feature vectors into a syllable-level sequence. It consists of: (i) a *differentiable soft-pooler* $m : \mathcal{X}^T \mapsto \mathcal{X}^L$ that aligns speech with text on the syllable level (ii) a tokenizer $c : \mathcal{X}^L \mapsto \tilde{\mathcal{X}}^L$ to discretizes speech into syllable-like units. Thus the speech pre-net is

$$e_{\tilde{X}}(\tilde{X}) = e_{\tilde{X}} \circ c \circ m(X), \tag{2}$$

where $f \circ g(x) := f(g(x))$ for any functions $f, g$, and $\tilde{X} := c \circ m(X) \in \tilde{\mathcal{X}}^L$. The soft-pooler first estimates the boundary probabilities $b(X) \in [0,1]^T$ by learning from an unsupervised syllable detector Sylber (Cho et al., 2025), then constructs a pooling mask $a(X) \in [0,1]^{L \times T}$, as illustrated in Figure 2:

$$\tilde{a}_{it}(X) := \sigma_{\epsilon}\left(i - \sum_{\tau \leq t} b_{\tau}(X)\right), \quad a_{it}(X) = \frac{\tilde{a}_{it}(X)}{\sum_{\tau=1}^{T} \tilde{a}_{i\tau}(X)}, \quad m_i(X) := \sum_{t=1}^{T} a_{it}(X) X_t, \tag{3}$$

where $\sigma_\epsilon(x) := \epsilon - |\mathrm{clamp}(x, -\epsilon, \epsilon)|$ and $\mathrm{clamp}(x, a, b) = x$ for $x \in [a, b]$, $a$ if $x < a$ and $b$ otherwise. The hyperparameter $\epsilon$ controls the *tapering speed* of pooling weights outside each syllabic segment. This creates sparse pooling weights that avoid overflow/underflow issues common in earlier soft-pooling methods with $\sigma_\epsilon(x) = \tanh(x/\epsilon)$ (Bhati et al., 2022) and $\sigma_\epsilon(x) = \exp(x/\epsilon)$ (Wang et al., 2024). To contextualize embeddings, both modalities are processed by a shared encoder $f : \mathbb{R}^{T \times d} \mapsto \mathbb{R}^{T \times d}$:

$$f_{\tilde{X}}(\tilde{X}) := f \circ e_{\tilde{X}}(\tilde{X}), \quad f_Y(Y) := f \circ e_Y(Y). \tag{4}$$

In practice, a differentiable K-means vector quantizer (Ni et al., 2025) is used for the tokenizer $c$ and a multi-layer transformer (Vaswani et al., 2017) is used for the shared encoder.

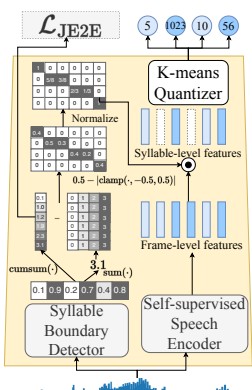

Figure 2: Speech syllabifier

**Distribution matching.** Without paired speech-text data, we approximate each unimodal distribution $p_{\tilde{X}}$ and $p_Y$. Let $g_{\tilde{X}}, g_Y : \mathbb{R}^{d \times L} \mapsto [0, 1]$ be speech and text *post-nets* such that

$$q_Z(Z) := g_Z \circ f_Z(Z) := \prod_{i=1}^{L} g_{Z,i}(Z_i | f_Z(Z^{\{1, \cdots, i-1\}})) = \prod_{i=1}^{L} \frac{e^{W_{iZ_i}^Z f_Z(Z^{\{1, \cdots, i-1\}})}}{\sum_z e^{W_{iz}^Z f_Z(Z^{\{1, \cdots, i-1\}})}}, \tag{5}$$

where $Z = \tilde{X}$ or $Y$ with equal probabilities is a *mixture random sequence*, and $Z_i^M := Z_i, i \in M$ and [MASK] otherwise for some index set $M$, and $W_i^Z \in \mathbb{R}^{\mathcal{Z} \times dL}$.

Training minimizes *Kullback–Leibler (KL) divergence* to unimodal distributions:

$$\min_{q_{\tilde{X}}, q_Y} D_{\mathrm{KL}}(p_{\tilde{X}} \| q_{\tilde{X}}) + D_{\mathrm{KL}}(p_Y \| q_Y) = D_{\mathrm{KL}}(p_{\tilde{X}} \| g_{\tilde{X}} \circ f_{\tilde{X}}) + D_{\mathrm{KL}}(p_Y \| g_Y \circ f_Y). \tag{6}$$

To prevent speech and text from occupying disjoint embedding regions, we further constrain the shared encoder *entropy*. Let $H(X)$ denotes the entropy of discrete random variable $X$, consider

$$\min_{f_{\tilde{X}}, g_{\tilde{X}}, f_Y, g_Y} D_{\mathrm{KL}}(p_{\tilde{X}} \| g_{\tilde{X}} \circ f_{\tilde{X}}) + D_{\mathrm{KL}}(p_Y \| g_Y \circ f_Y) \quad \text{s.t.} \quad H(f_Z(Z)) \leq H(Y), \tag{7}$$

**Theoretical guarantee.** We prove that under regularity conditions, equation 7 matches true and generated text distributions in the same way as GANs, but without unstable straight-through gradients. Thus zero-error UASR is achievable under conditions in (Wang et al., 2023b).

**Theorem 1.** *Suppose $(f_{\tilde{X}}^*, g_{\tilde{X}}^*, f_Y^*, g_Y^*)$ minimize equation 7, then under certain assumptions (See Appendix B), $f_{\tilde{X}}^*$ and $f_Y^*$ are invertible and $q_{Y|X}^*(y|x) := \mathbb{1}[f_Y^{*-1} \circ f_{\tilde{X}}^* \circ c \circ m(x) = y]$ satisfies*

$$D_{\mathrm{KL}}(p_Y \| \mathbb{E}_X[q_{Y|X}^*]) = 0, \quad y^*(x) = \underset{y \in \mathcal{Y}^L}{\mathrm{argmax}}\, q_{Y|X}^*(y|x), \quad \forall x \in \mathcal{X}^T.$$

**Practical training.** In practice, SylCipher implements unimodal masked language modeling (MLM) to approximate unimodal probability distributions. Given a mask distribution $p_M$, we minimize the weighted loss:

$$\mathcal{L}_{\mathrm{wMLM}}(\theta_X, \theta_Y) := -\mathbb{E}_{Z \sim \frac{1}{2} p_{\tilde{X}} + \frac{1}{2} p_Y, M \sim p_M} \left[ \ln q_Z(Z^M | Z^{M^c}) + \lambda \ln q_Z(Z^M | Z^M) \right], \tag{8}$$

where $\theta_X$ and $\theta_Y$ are speech-related and text-related parameters. Notice that $g_Y$ serves a dual role: it acts both as the text post-net for unimodal MLM and as a *cross-modal decoder*, defined as (similar for $q_X$) $q_Y(Y^M | Z^M) := \prod_{i \in M} g_{Y,i}(Y_i | f_Z(Z_i))$. The additional term weighted by $\lambda$ balances these two functions, enabling $g_Y$ (or $g_X$) to reconstruct text (or speech) tokens from either masked text or speech inputs. To further constrain the entropy of the encoding outputs $f_Z(Z)$, we limit the transformer depth to two layers and apply *random mix-up* (Ao et al., 2022). During training, a random subset of encoder representations is quantized to a finite set of code vectors before passed to the post-nets. This restricts the variability of $f_Z(Z)$ without significantly reducing its predictive capacity for masked tokens.

After pretraining with unsupervised syllable boundary labels $[\hat{b}_1(X), \cdots, \hat{b}_T(X)]$, we perform *joint end-to-end* (JE2E) training. In this stage, the soft-pooler is trained jointly with the rest of the model, guided by an

additional soft constraint on the predicted *syllable counts*:

$$\mathcal{L}_{\text{JE2E}}(\theta_X) := \mathbb{E}_X \left| \sum_{t=1}^{T} b_t(X) - \hat{b}_t(X) \right|, \tag{9}$$

which discourages both over- and under-segmentation.

While the MLM objective encourages *implicit* distribution matching, we found that *explicit* distribution matching further improves a saturated MLM system. To this end, we adopt positional unigram and skipgram matching (PUSM) (Wang et al., 2024; Ni et al., 2025):

$$\mathcal{L}_{\text{PUSM}}(\theta_X, \theta_Y) = \left\| \mathbb{E}_X \left[ q_{Y_i}(X) \right] - p_{Y_i} \right\|_1 + \sum_{k=1}^{K} \left\| \mathbb{E}_X \left[ q_{Y_i Y_{i+k}}(X) \right] - p_{Y_i Y_{i+k}} \right\|_1, \tag{10}$$

where $q_Y(x) := q_Y(\cdot | c \circ m(x))$ and $K$ is the maximal skip length. The first term matches unigram distributions at each position, while the second aligns skipgram distributions. To stabilize training, we disable random mix-up and approximate the empirical probability distributions using the entire dataset as a single batch, reducing memory cost via gradient accumulation (Ni et al., 2025). The overall SylCipher objective combines all stages:

$$\mathcal{L}_{\text{SylCipher}} := \lambda_{\text{wMLM}} \mathcal{L}_{\text{wMLM}} + \lambda_{\text{JE2E}} \mathcal{L}_{\text{JE2E}} + \lambda_{\text{PUSM}} \mathcal{L}_{\text{PUSM}}. \tag{11}$$

Because the PUSM stage requires different batch sizes, we adopt an *iterative training schedule* rather than fully end-to-end optimization. Specifically:

1. **Fixed boundary stage**: freeze the boundary detector $b$ and set $\lambda_{\text{JE2E}} = \lambda_{\text{PUSM}} = 0$;
2. **JE2E stage**: enable $\lambda_{\text{JE2E}} > 0$ to refine segmentation;
3. **PUSM stage**: disable $\lambda_{\text{wMLM}}$, enable $\lambda_{\text{PUSM}} > 0$ for explicit distribution matching.

### 4.2 INFERENCE

During inference, the ASR system cascades the speech syllabifier, speech pre-net and the text post-net as done in the PUSM stage:

$$y(X) := \underset{y \in \mathcal{Y}^L}{\arg\max} \, q_Y(y|X). \tag{12}$$

We observed that using the *first* transformer layer of the shared encoder, instead of the last, improves performance, consistent with findings at the word-level (Ni et al., 2025). This may be due to over-contextualization in later layers. Finally, replacing each <OOV> token with the second most likely prediction reduces CER by about $1\%$ compared to simply discarding <OOV>s.

## 5 EXPERIMENTS

Section 5.1 introduces the datasets used for our experiments, followed by the syllabification steps for speech and text preprocessing in Section 5.2. Section 5.3 presents the main UASR results, and Section 5.4 discusses SylCipher's boundary refinement ability. Section 5.5 provides ablation studies on key design choices. Additional implementation details of our method and the baselines are given in Appendix D.

### 5.1 DATASETS

We evaluate SylCipher on three datasets. First, we train on the 460-hour clean subset of LibriSpeech (Panayotov et al., 2015a), a standard UASR benchmark of audiobook recordings. Second, to test domain generalization, we train another model on SpokenCOCO (Hsu et al., 2021b), which contains 742 hours of spoken image captions. Third, to study languages with syllabic structures significantly different from English, we apply SylCipher to Mandarin using AISHELL-3 (Shi et al., 2021), which has 85 hours of read speech. We follow the same LibriSpeech split as in (Ni et al., 2025), and use the standard splits for SpokenCOCO and AISHELL-3. For LibriSpeech and SpokenCOCO, we consider both the *matched* setting, where empirical speech and text probability distributions can be matched exactly, and the more realistic *unmatched* setting where they cannot. In the matched case, we use paired speech-text datasets with pairings removed; in the unmatched case, LibriSpeech uses LibriLM (Panayotov et al., 2015b) with overlapping text removed, while SpokenCOCO is randomly split in half, with one half treated as speech-only and the other half treated as text-only. For AISHELL-3, we consider only the matched setting and compare models trained with/without tone labels. For all datasets, we apply a voice activity detector[1] to improve alignment.

---

[1] https://github.com/wiseman/py-webrtcvad.git

Table 1: **UASR results on LibriSpeech (clean subsets) and SpokenCOCO**. Inside the bracket lists the unsupervised boundary used for each model. *Student* stands for the student model used during the self-training stage using pseudo-labels from each model. For tokens used for the text data, *Char.* stands for characters and *Syllable* stands for syllable-level tokens converted using the Pyphen+ syllabifier without using a G2P. CER stands for character error rate. * indicates evaluation on LibriSpeech dev-clean instead.

(a) UASR results on LibriSpeech (clean subsets)

| Model | Student | Token | Matched CER (↓) | Unmatched CER (↓) |
|---|---|---|---|---|
| *G2P-based approach* | | | | |
| wav2vec-U* (Baevski et al., 2021) | None | Phone | - | 13.3 |
| wav2vec-U 2.0* (Liu et al., 2023) | None | Phone | - | 12.2 |
| REBORN* (Tseng et al., 2024) | None | Phone | - | 8.3 |
| *G2P-free approach* | | | | |
| wav2vec-U (Baevski et al., 2021) | None | Char. | 35.6 | 43.3 |
| | wav2vec 2.0 | Char. | 33.8 | 42.1 |
| REBORN (Tseng et al., 2024) | None | Char. | 37.8 | 76.6 |
| JSTTI (forced align) | None | Char | 81.4 | 81.1 |
| JSTTI (Ni et al., 2025) | None | Word | 49.5 | 54.2 |
| PUSM (Sylber) (Wang et al., 2023c) | None | Syllable | 35.5 | 57.7 |
| | wav2vec 2.0 | Syllable | 33.0 | 54.7 |
| SylCipher (Ours, forced align) | None | Syllable | 38.5 | 46.4 |
| SylCipher (Ours, Sylber) | None | Syllable | 43.5 | 48.6 |
| SylCipher (Ours, Sylber+JE2E) | None | Syllable | 39.2 | 46.8 |
| SylCipher (Ours, Sylber+JE2E+PUSM) | None | Syllable | **21.8** | **35.9** |
| | wav2vec 2.0 | Syllable | **17.5** | **33.3** |

(b) UASR results on SpokenCOCO

| Model | Student | Token | Matched CER (↓) | Unmatched CER (↓) |
|---|---|---|---|---|
| wav2vec-U (Baevski et al., 2021) | None | Char. | 45.0 | 45.2 |
| | wav2vec 2.0 | Char. | 46.3 | 35.3 |
| JSTTI | None | Char. | 78.3 | 100 |
| JSTTI (Ni et al., 2025) | None | Word | 64.5 | 64.5 |
| PUSM (Sylber) (Wang et al., 2023c) | None | Syllable | 41.5 | 41.3 |
| | wav2vec 2.0 | Syllable | 34.7 | 34.3 |
| SylCipher (Ours, Sylber) | None | Syllable | 34.9 | 36.1 |
| SylCipher (Ours, Sylber+JE2E) | None | Syllable | 31.2 | 32.4 |
| SylCipher (Ours, Sylber+JE2E+PUSM) | None | Syllable | **23.4** | **26.8** |
| | wav2vec 2.0 | Syllable | **13.9** | **17.6** |

## 5.2 SPEECH AND TEXT SYLLABIFICATION

For English text, we use Pyphen[2], a rule-based hyphenation tool re-purposed for syllabification without a G2P. If Pyphen produces only a single chunk for a long word, we apply a simple rule-based fallback (Appendix E). We denote this combined approach as Pyphen+. We also experiment with other G2P-free approaches such as byte-pair encoding (BPE) Liu et al. (2025); Sennrich et al. (2016), and find SylCipher robust to syllabification errors. To avoid long-tail distributions, we keep only the top-2048 most frequent English syllables and replace the rest with a special <OOV> token (replacing 7% of tokens in LibriSpeech and 2% in SpokenCOCO. For Mandarin, we use the Pinyin of each Chinese character as a syllable, with or without tone labels. We keep the top-1024 most frequent syllables, covering 99.5% of occurrences. For speech syllabification, we use K-means clustering on syllable-level speech features created by mean pooling within the Sylber boundaries, with codebook size equal to the number of non-<OOV> text tokens.

## 5.3 RESULTS: UASR

We compare SylCipher with UASR systems that differ in token type, training objective, and architecture.

---

[2]https://github.com/Kozea/Pyphen

Table 2: **UASR results on AISHELL-3 test set**. Inside the bracket lists the unsupervised boundary used for each model. "Student" stands for the student model used during the self-training stage using pseudo-labels from each model. *Init./Final* refers to initials and finals in the Chinese phonetic alphabet. For text data tokens, *Syllable* refers to the pinyin representation of each Chinese character, while *Phone* denotes the individual letters within the pinyin tokens. PER stands for phone error rate.

| Model | Student | Token | w/o Tone PER ($\downarrow$) | w. Tone PER ($\downarrow$) |
|---|---|---|---|---|
| wav2vec-U (Baevski et al., 2021) | None | Phone | 74.9 | 76.2 |
| JSTTI | None | Init./Final | 96.4 | 100 |
| JSTTI (Ni et al., 2025) | None | Word | 83.2 | 169 |
| PUSM (Sylber) (Wang et al., 2023c) | None | Syllable | 28.4 | 26.5 |
|  | wav2vec 2.0 | Syllable | 18.5 | 14.9 |
| SylCipher (Ours, forced align) | None | Syllable | 38.1 | 38.9 |
| SylCipher (Ours, Sylber) | None | Syllable | 44.6 | 48.3 |
| SylCipher (Ours, Sylber+JE2E) | None | Syllable | 41.7 | 45.1 |
| SylCipher (Ours, Sylber+JE2E+PUSM) | None | Syllable | **26.9** | **24.9** |
|  | wav2vec 2.0 | Syllable | **15.3** | **12.2** |

**Word-level:** *JSTTI* (Ni et al., 2025), the state-of-art word-level UASR system, which is GAN-free and architecturally similar to ours, using 2048 non-<OOV> words.

**Character-level:** *wav2vec-U* (Baevski et al., 2021), a strong GAN-based system; *REBORN* (Tseng et al., 2024), a state-of-the-art phoneme-based GAN system; and a *phone-level JSTTI*, trained with phoneme boundaries and 128 speech clusters (comparable to the number of character types).

**Syllable-level:** *PUSM* (Wang et al., 2023c), adapted from word-level UASR to syllables using the same syllable boundary detector and syllabifier as SylCipher.

We also report results with *self-training* Chen et al. (2019); Baevski et al. (2021), where a wav2vec 2.0 Baevski et al. (2020) student is further finetuned by distilling character (grapheme)-level pseudo-labels from a UASR system. Performance is measured by character error rate (CER) for English and phone error rate (PER) for Mandarin, as both are tokenization-independent and easily comparable.

**Syllable-level modeling performs best under G2P-free setting.** Table 1a summarizes results on LibriSpeech. Among baselines, PUSM performs best in the matched setting, while wav2vec-U is strongest in the unmatched setting, though its performance is limited by the lack of G2P. Unlike phonemized training, REBORN trained directly on raw characters performs worse than wav2vec-U, suggesting sensitivity to misalignment between speech and text. By contrast, SylCipher with all three stages (Sylber+JE2E+PUSM) achieves 21.8% CER (matched) and 35.9% (unmatched), outperforming all baselines. Compared to the best-in-average system wav2vec-U (35.6%/43.3%), SylCipher reduces by CER by 40% (matched) and 17% (unmatched) relative. Both syllable-level models (PUSM and SylCipher) outperform word- or character-level models, confirming that speech-text alignment is the best at the syllable level. Word-level JSTTI performs worst due to poor rare-word coverage (<OOV>s ≈ 17%), and adapting JSTTI to phone-level degrades further, likely because phone clusters are noisier than syllable or word clusters.

**Iterative training helps.** Stage-wise training shows progressive improvements: JE2E reduces CER modestly, while PUSM yields the largest gains (44% and 23% relative reductions over JE2E in matched/unmatched settings). Combining MLM-based stages with PUSM outperforms PUSM-only training by 33-34% relative, as MLM provides necessary initialization for PUSM to converge. Indeed, PUSM alone fails when SylCipher is randomly initialized, likely due to transformer training instability. Lastly, using unsupervised Sylber boundaries performs nearly as well as forced alignment, suggesting robustness to segmentation noise. Self-training consistently improves performance, especially for stronger models. For SylCipher, CER is further reduced by 20% relative.

**Syllables are robust to domain shifts.** Table 1b reports results on SpokenCOCO. SylCipher again outperforms all baselines, surpassing PUSM by 32% and wav2vec-U by 49% relative CER after all stages. The margin is larger than LibriSpeech, especially in the unmatched setting. Notably, even after the first training stages, SylCipher already outperforms wav2vec-U by 22%. While wav2vec-U suffers from domain mismatch at the character level, both syllable-level methods perform better, suggesting syllable units are more robust to domain shifts. Self-training further improves SylCipher by 39% relative CER, likely due to

higher syllable coverage reducing insertion/deletion errors. Moreover, performance degrades only slightly in the unmatched setting, indicating the sharper degradation on LibriSpeech stems from domain mismatch between its speech and text corpora.

**Syllable-level UASR works for Mandarin.** Table 2 presents results on Mandarin. Here, syllable-level models converge even without boundary refinement, while phone- and word-level approaches struggle, confirming that syllables are the most natural alignment unit for Mandarin. Compared to the best baseline (PUSM), SylCipher achieves over 5.5% relative PER reduction before self-training and 17% after. Interestingly, including tone labels does not harm performance–in fact, PER improves after the PUSM stage–suggesting that once phoneme labels are predicted correctly, tone prediction is also reliable.

Table 3: **Syllable boundary detection results on LibriSpeech and SpokenCOCO**. The superscript for F1 scores is tolerance threshold in ms and the tolerance is 50ms for other metrics. P., Re. and R stand for precision, recall and R-value respectively. SylCipher is trained under unmatched settings.

| (a) Boundary detection results on LibriSpeech | | | | | |
|---|---|---|---|---|---|
| | **F1**$^{50}$ (↑) | **F1**$^{20}$ (↑) | **P.** (↑) | **Re.** (↑) | **R** (↑) |
| Feat-Sim (Peng et al., 2023) | 47.3 | 24.7 | 46.6 | 48.0 | 54.4 |
| SDHuBERT (Cho et al., 2024) | 66.1 | 32.2 | 64.9 | 67.4 | 70.7 |
| SylBoost (Baade et al., 2025) | 73.2 | 44.6 | 72.1 | 74.4 | 76.9 |
| Sylber (Cho et al., 2025) | 83.4 | 44.1 | 84.8 | 84.1 | 86.4 |
| SylCipher (Ours, Sylber+JE2E) | **86.1** | **50.8** | **86.6** | **86.1** | **88.1** |
| (b) Boundary detection results on SpokenCOCO | | | | | |
| | **F1**$^{50}$ (↑) | **F1**$^{20}$ (↑) | **P.** (↑) | **Re.** (↑) | **R** (↑) |
| Feat-Sim (Peng et al., 2023) | 60.3 | - | 57.4 | 63.6 | 64.3 |
| SylBoost (Baade et al., 2025) | 55.6 | 30.2 | 48 | **65.2** | 50.8 |
| Sylber (Cho et al., 2025) | 53.5 | 28.6 | 52.3 | 54.9 | 59.6 |
| SylCipher (Ours, Sylber+JE2E) | **62.3** | **31.0** | **61.1** | 63.6 | **67.4** |

## 5.4 RESULTS: UNSUPERVISED SYLLABLE BOUNDARY DETECTION

To better understand the JE2E stage, we compare the syllable boundary detection performance against the teacher model Sylber (Cho et al., 2025) (which provides SylCipher's initial boundaries) and other unsupervised approaches including Feat-Sim (Peng et al., 2023), SDHuBERT (Cho et al., 2024), Syl-Boost (Baade et al., 2025). On LibriSpeech, Sylber is the strongest baseline on most metrics, except for the 20ms F1 score where SylBoost performs best. On SpokenCOCO, SylBoost outperforms Sylber on three of five metrics. However, our preliminary analysis show that SylBoost often over-segments, producing too many syllables and causing cross-modal misalignment and training instability in SylCipher. In contrast, Sylber predicts syllable counts closer to ground truth, making it more suitable as an initialization for UASR. When refined through JE2E, SylCipher improves upon its Sylber teacher, achieving +14% relative F1 (20ms tolerance) and +3% relative F1 (50ms) on LibriSpeech. On SpokenCOCO, it surpasses Sylber by +15% F1 and +11% R-value, and outperforms the best baseline Feat-Sim by +3% relative F1 score and +4.6% R-value. These results suggest that unpaired text provides additional guidance on syllable boundary detection. Visualizations of the speech-text alignment predicted by SylCipher can be found in Appendix G.

## 5.5 ABLATION STUDIES

We conduct ablation studies on various components and design choices in SylCipher in Figure 3.

**SylCipher is robust to syllabifiers.** We first test alternative syllabifiers beyond Pyphen, including the phoneme-based Syllabify[3] and a character-based approach using byte-pair encoding (BPE) tokenization Sennrich et al. (2016). The latter is attractive because it is language-agnostic and integrated easily with spoken language models. For the BPE-based syllabifier, we train the first stage of SuperBPE Liu et al. (2025) on LibriSpeech with a 17k vocabulary, roughly matching the number of syllable types. We then split BPE tokens containing multiple non-consecutive vowels and merge consecutive tokens to ensure each unit contains at least one vowel (details in Appendix F). We refer to this method as BPE+. All syllabifier variants are trained only under the *fixed-boundary stage*, since later stages show correlated trends. Instead of CER, we report syllable error rate (SER) to more directly reflects syllable-level performance. Among

---

[3]https://github.com/kylebgorman/syllabify

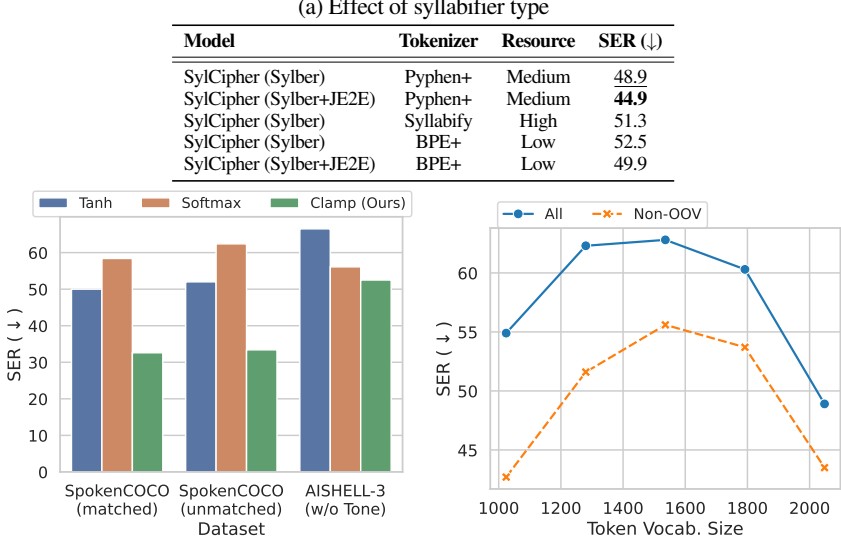

(a) Effect of syllabifier type

| Model | Tokenizer | Resource | SER (↓) |
|---|---|---|---|
| SylCipher (Sylber) | Pyphen+ | Medium | 48.9 |
| SylCipher (Sylber+JE2E) | Pyphen+ | Medium | **44.9** |
| SylCipher (Sylber) | Syllabify | High | 51.3 |
| SylCipher (Sylber) | BPE+ | Low | 52.5 |
| SylCipher (Sylber+JE2E) | BPE+ | Low | 49.9 |

(b) SER vs. pooler type across datasets     (c) SER vs. token vocabulary size

Figure 3: **Ablation studies on the effect of syllabifier type, pooling type and token vocabulary size on SylCipher UASR performance**. (a) The effect of different pooler named after the type of $\sigma_\epsilon$ function used in equation 3.(b) The effect of the vocabulary size of non-<OOV>-tokens on SylCipher performance during the fixed-boundary stage on LibriSpeech (matched). (c) Effect of syllabifiers with difference resource requirements on SylCipher performance during the fixed-boundary stage on LibriSpeech (matched).

the methods, Pyphen+ achieves the lowest SER, outperforming even the phoneme-based Syllabify. As shown in Figure 3a, BPE+ yields weaker but still competitive results, demonstrating that SylCipher can generalize to linguistically simpler, resource-free syllabifiers.

**Clamp-based soft-pooler trains more stably.** In Figure 3b, we test our modified soft-pooler in equation 3, which uses a clamp-based $\sigma_\epsilon$ function ("Clamp"), against alternatives based on $\tanh$ (Bhati et al., 2022) ("Tanh") and softmax (Wang et al., 2024)) ("Softmax"). Following prior work, we set the tapering parameter $\epsilon = 0.5$ for "Clamp" and $0.1$ for the other two approaches. On both SpokenCOCO and AISHELL-3, Clamp consistently outperforms the sigmoid-based poolers by 6-36%. This suggests that our design provides more stable and efficient training dynamics.

**SylCipher works with different vocabulary sizes.** We also experiment with varying the non-<OOV> vocabulary size in Figure 3c. SylCipher displays consistent SERs across a wide range of vocabulary sizes, reaching the lowest overall SER with 2048 tokens and the lowest non-<OOV> SER with 1024 tokens.

## 6 CONCLUSION

In this work, we introduced SylCipher, a UASR system that avoids phoneme-level resources such as G2P by recognizing speech at the *syllable level*. Under the G2P-free setting, SylCipher outperforms the best existing systems by 17-40% relative CER on LibriSpeech and shows generalizability across other domains on SpokenCOCO, narrowing the gap with G2P-based systems. It also demonstrates cross-lingual robustness: on Mandarin, a tonal language where phoneme-based methods fail to converge, SylCipher achieves a PER of 13%. These results suggest that syllable-level modeling is a viable alternative to phoneme-level UASR and can push the horizon of more accessible and inclusive spoken language technology.

**Limitations.** SylCipher is not yet language-universal, since different languages use different writing systems and require linguistic knowledge to properly syllabify. For example, languages such as Hebrew and Arabic omits vowels in their writing system, which could pose challenges to existing syllabifiers. While our experiments show that the method can be adapted to more resource-efficient tokenizers such as BPE with minimal modifications, coming up with a language-universal tokenization method remains an open problem. Further, the iterative training procedure can be further simplified into an end-to-end approach. Lastly, improving the robustness of SylCipher under domain mismatch between speech and text remains an open challenge.

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

## A    LIMITATIONS

SylCipher is not yet language-universal, since different languages use different writing systems and require linguistic knowledge to properly syllabify. For example, languages such as Hebrew and Arabic omits vowels in their writing system, which could pose challenges to existing syllabifiers. While our experiments show that the method can be adapted to more resource-efficient tokenizers such as BPE with minimal modifications, coming up with a language-universal tokenization method remains an open problem. Further, the iterative training procedure can be further simplified into an end-to-end approach. Lastly, improving the robustness of SylCipher under domain mismatch between speech and text remains an open challenge.

## B    PROOF OF THEOREM 1

We state the full version of Theorem 1 here.

**Theorem 2.** *Let $(f_{\tilde{X}}^*, g_{\tilde{X}}^*, f_Y^*, g_Y^*)$ be a minimizer of equation 7, and suppose the following assumptions hold:*

1. *The speech feature sequence $X$ is a sequence of one-hot vectors with $|\mathcal{X}| = |\tilde{\mathcal{X}}| = |\mathcal{Y}|$;*

2. *The true syllable boundaries are used, i.e., $b_t(X) = \mathbb{1}[y^*(X_t) \neq y^*(X_{t+1})]$;*

3. *The tokenizer $c$ is an optimal K-means quantizer with Euclidean distance metric and cluster size $|\tilde{\mathcal{X}}|$;*

4. *The true ASR $y^*$ is decomposable, i.e., $y^*(\tilde{X}) = [y^*(\tilde{X}_1), \cdots, y^*(\tilde{X}_L)]$;*

5. *$f_{\tilde{X}}$ and $f_Y$ are decomposable;*

6. *Assumption 1-2 in (Wang et al., 2023b) hold for $(\tilde{X}, Y)$.*

*Then $f_{\tilde{X}}^*$ and $f_Y^*$ are invertible and $q_{Y|X}^*(y|x) := \mathbb{1}[f_Y^{*-1} \circ f_{\tilde{X}}^* \circ c \circ m(x) = y]$ satisfies*

$$D_{\mathrm{KL}}(p_Y || \mathbb{E}_X[q_{Y|X}^*]) = 0, \quad y^*(x) = \operatorname*{argmax}_{y \in \mathcal{Y}^L} q_{Y|X}^*(y|x), \quad \forall x \in \mathcal{X}^T.$$

The proof relies on the following lemma proven in Appendix C.

**Lemma 1.** *Suppose Assumption 5 of Theorem 1 and Assumption 1 and 2 of (Wang et al., 2023b) hold for $(\tilde{X}, Y)$, and functions $g_X : \tilde{\mathcal{X}}^L \mapsto [0,1], g_Y : \tilde{\mathcal{Y}}^L \mapsto [0,1]$ of the form in equation 5 be such that $p_X = g_X \circ f_X$ and $p_Y = g_Y \circ f_Y$. Then $f_X$ and $f_Y$ are both invertible.*

Now we are ready to prove the main theorem.

*Proof.* The proof consists of two main parts: (i) First, we prove that at least one minimizer $(f_{\tilde{X},1}, g_{\tilde{X},1}, f_{Y,1}, g_{Y,1})$ can achieve a minimum of 0 for equation 7; (ii) then we establish that for any minimizers $(f_{\tilde{X}}^*, g_{\tilde{X}}^*, f_Y^*, g_Y^*)$, the marginal distribution of the text posterior $q_{Y|X}^*$ matches the true text distribution $p_Y$. As a result, by Assumption 4, Theorem 1 of (Wang et al., 2023b) guarantees that $\operatorname{argmax}_{y \in \mathcal{Y}^L} q_{Y|X}^*(y|X)$ achieves zero-error UASR.

To prove (i), by the definition of $y^*$ and Assumption 1, $y^*$ is a invertible mapping between $\mathcal{X}$ and $\mathcal{Y}$. Then let $t_i = \min\{\tau : \sum_{t=1}^{\tau} b_t(X) \geq i-1\}$ be the *starting time* of each syllable and apply Assumption 2, we have

$$X_s = X_t, \forall t_i \leq s < t < t_{i+1}, 1 \leq i \leq L.$$

As a result, the output of the soft-pooler is simply $m_i(X) = X_{t_i}$. Further, by Assumption 3, $c$ is optimal, and we claim that this is achievable if and only if for any $(x, x') \in \mathcal{X}^2$,

$$c(x) = c(x') \Longleftrightarrow y^*(x) = y^*(x').$$

Otherwise, suppose for some $(x, x') \in \mathcal{X}^2$ such that $c(x) = c(x') =: c_0$ but $y^*(x) \neq y^*(x')$, then for any centroid $\mu_{c_0}$ of cluster $c_0$ and triangle inequality,

$$\|x - \mu_{c(x)}\|_2 + \|x' - \mu_{c(x')}\|_2 = \|x - \mu_{c_0}\|_2 + \|x' - \mu_{c_0}\|_2 \geq \|x - x'\|_2 > 0,$$

and therefore the K-means objective:

$$\mathcal{L}_{\mathrm{km}}(\mu_1,\cdots,\mu_{|\mathcal{X}|}):=\mathbb{E}_{X\sim p_X}\|X-\mu_{c(X)}\|_2^2>0.$$

However, by simply setting $c(x)=y^*(x)$ for all $x\in\mathcal{X}$, we have $\mu_{c(x)}=x$ for any $x\in\mathcal{X}$ and $\mathcal{L}_{\mathrm{km}}=0$ due to the invertibility of $y^*$. This contradicts with the optimality of $c$ and thus proves the "$\Longrightarrow$" direction. Further, this implies that the cluster size is at least $|\mathcal{Y}|$ and since we set the cluster size to $|\mathcal{Y}|$, $c\circ m$ is equivalent to $y^*$ up to a permutation. This then proves the other direction. Let $c\circ m=\pi\circ y^*$ where $\pi$ is a permutation, then the syllable-level features $\tilde{X}=\pi\circ y^*(X)$ and thus by the permutation-invariance of discrete entropy,

$$H(\tilde{X})=H(y^*(X))=H(Y).$$

Therefore, set $f_{\tilde{X},1}$ to be a permutation on $\tilde{\mathcal{X}}$, $f_{Y,1}=f_{\tilde{X},1}\circ\pi$, $g_{\tilde{X},1}=p_{\tilde{X}}\circ f_{\tilde{X},1}^{-1}$ and $g_{Y,1}=p_Y\circ f_{Y,1}^{-1}$, we have

$$D_{\mathrm{KL}}(p_{\tilde{X}}\|g_{\tilde{X},1}\circ f_{\tilde{X},1})+D_{\mathrm{KL}}(p_Y\|g_{Y,1}\circ f_{Y,1})=D_{\mathrm{KL}}(p_{\tilde{X}}\|p_{\tilde{X}})+D_{\mathrm{KL}}(p_Y\|p_Y)=0,$$

and

$$f_{Y,1}(y(X))=f_{\tilde{X},1}\circ\pi\circ y(X)=f_{\tilde{X},1}\circ c\circ m(X)=f_{\tilde{X},1}(\tilde{X}).$$

The latter implies that the probability distribution of $f_{Z,1}(Z)$ satisfies

$$p_{f_{Z,1}(Z)}=\frac{1}{2}p_{f_{\tilde{X},1}(\tilde{X})}+\frac{1}{2}p_{f_{Y,1}(Y)}=\frac{1}{2}p_{f_{\tilde{X},1}(\tilde{X})}+\frac{1}{2}p_{f_Y(y(X)),1}=p_{f_{\tilde{X},1}(\tilde{X})}, \tag{*}$$

and thus

$$H(f_{Z,1}(Z))=H(f_{Y,1}(Y))=H(f_{X,1}(\tilde{X}))\le H(Y).$$

Therefore, by the nonnegativity of $D_{\mathrm{KL}}$, $(f_{\tilde{X},1},g_{\tilde{X},1},f_{Y,1},g_{Y,1})$ is an optimal solution of equation 7 with a minimum of 0, which proves $(i)$.

To prove (ii), we first use (i) to conclude that

$$p_{\tilde{X}}=g_{\tilde{X}}^*\circ f_{\tilde{X}}^*,p_Y=g_Y^*\circ f_Y^*.$$

Then it amounts to prove that any minimizer $(f_{\tilde{X}}^*,g_{\tilde{X}}^*,f_Y^*,g_Y^*)$ of equation 7 satisfies

$$p_{f_Z^*(Z)}=p_{f_X^*(X)}=p_{f_Y^*(Y)}. \tag{*}$$

Since if this is the case, by Lemma 1, $f_Y^*$ is invertible and thus for any $y\in\mathcal{Y}^L$,

$$p_{f_Y^{*-1}\circ f_X^*(X)}(y)\overset{(a)}{=}p_{f_X^*(X)}(f_Y^*(y))\overset{(b)}{=}p_{f_Y^*(Y)}(f_Y^*(y))\overset{(c)}{=}p_Y(y)$$
$$\Longrightarrow D_{\mathrm{KL}}(p_Y\|\mathbb{E}_X q_{Y|X}^*)=0,$$

where $(b)$ uses (*) and $(a)(c)$ uses Lemma 1.

To prove (*), we use the concavity of the discrete entropy $h(p):=-\sum_x p(x)\log p(x)$ and Lemma 1,

$$H(f_Z^*(Z))=h\left(\frac{1}{2}p_{f_X^*(X)}+\frac{1}{2}p_{f_Y^*(Y)}\right)\overset{(d)}{\ge}\frac{1}{2}h(p_{f_X^*(X)})+\frac{1}{2}h(p_{f_Y^*(Y)})$$
$$\overset{(e)}{=}\frac{1}{2}(H(X)+H(Y))=H(Y),$$

with equality if and only if $p_{f_Y^*(Y)}=p_{f_X^*(X)}$, where $(d)$ uses the concavity of the entropy function and $(e)$ uses Lemma 1. This concludes the proof of $(ii)$. $\qquad\square$

## C  PROOF OF LEMMA 1

*Proof.* We prove the lemma by contradiction and focus on proving the invertibility of $f_X$ since the proof is analogous for $f_Y$. Suppose otherwise $f_X$ is not invertible, then by Assumption 5, there exists $(x',x'')\in\tilde{\mathcal{X}}^2$ such that $f_X(x')=f_X(x'')$ but $x'\neq x''$, then by the definition of $g_X$,

$$p_{X_i}(x')=\sum_{x_{-i}\in\tilde{\mathcal{X}}^{L-1}}p_X(x_1,\cdots,x_{i-1},x',x_{i+1},\cdots,x_L)$$

$$=\sum_{x_{-i}\in\mathcal{X}^{L-1}}\prod_{j<i}g_X(x_j|f_X(x_1),\cdots,f_X(x_{j-1}))\cdot$$

$$\prod_{k\geq i}g_X(x_k|f_X(x_1),\cdots,f_X(x_{i-1}),f_X(x'),f_X(x_{i+1})\cdots,f_X(x_{k-1}))$$

$$=\sum_{x_{-i}\in\mathcal{X}^{L-1}}\prod_{j<i}g_X(x_j|f_X(x_1),\cdots,f_X(x_{j-1}))\cdot$$

$$\prod_{k\geq i}g_X(x_k|f_X(x_1),\cdots,f_X(x_{i-1}),f_X(x''),f_X(x_{i+1})\cdots,f_X(x_{k-1}))$$

$$=\sum_{x_{-i}\in\tilde{\mathcal{X}}^{L-1}}p_X(x_1,\cdots,x_{i-1},x'',x_{i+1},\cdots,x_L)=p_{X_i}(x''). \quad (13)$$

Therefore, the positional unigram matrix

$$P^X:=\begin{bmatrix}p_{X_1}^\top\\\vdots\\p_{X_L}^\top\end{bmatrix}$$

is column-rank-deficient. However, Theorem 1 of (Wang et al., 2023b) asserts that if Assumption 1 and 2 of (Wang et al., 2023b) holds, $P^X$ has full column-rank, which is a contradiction. Therefore, $f_X(x')\neq f_X(x'')$ if $x'\neq x''$ and $f_X$ is invertible. □

## D  IMPLEMENTATION DETAILS OF SYLCIPHER

For English experiments, SylCipher uses a HuBERT-large[4] model pretrained on LibriLight (Kahn et al., 2020) as the SSL encoder in the speech syllabifier, and initialize the soft-pooler with unsupervised boundary labels from Sylber (Cho et al., 2025). For Mandarin, we instead use XEUS[5] (Chen et al., 2024), which is pretrained on Mandarin (among other languages) and significantly outperforms HuBERT. We also finetune Sylber on AISHELL-3 to ensure stable convergence. For the pre-nets, shared encoder, post-nets, MLM parameters, and most of the optimizer hyperparameters, we follow JSTTI (Ni et al., 2025), as modifying them did not yield consistent improvements.

---

[4]https://dl.fbaipublicfiles.com/hubert/hubert_large_ll60k.pt
[5]https://huggingface.co/espnet/xeus/blob/main/model/xeus_checkpoint_new.pth

# E   PSEUDO-CODE FOR THE PYPHEN+ SYLLABIFIER

---

**Algorithm 1** Merging No-Vowel Segments and Naive Syllabification

---

```
1:  function MERGE_NOVOWEL_SEGMENTS(segs)
2:      new_segs ← [ ]
3:      n_seg ← len(segs)
4:      i ← 0
5:      while i < n_seg do
6:          seg ← segs[i]
7:          if seg has no vowels then
8:              if seg is the last segment then
9:                  merge seg into previous segment (if any)
10:             else
11:                 merge seg into the next segment
12:                 i ← i+1
13:             end if
14:         else
15:             append seg to new_segs
16:         end if
17:         i ← i+1
18:     end while
19:     return new_segs
20: end function
21: function NAIVE_SYLLABIFY(w)
22:     w0 ← w
23:     if w ends with a silent 'e' (not "-le") then
24:         drop the final 'e' in w
25:     end if
26:     if w ends with "-ed" and root does not end with 't' or 'd' then
27:         drop the 'e' in "-ed" in w
28:     end if
29:     Apply rule to w: split V C-CC...  V into VC-CC...V
30:     Apply rule to w: split VCCV into VC-CV
31:     Apply rule to w: split VCV into V-CV
32:     if modifications were made then
33:         if w0 ended with 'e' then
34:             add back 'e' to w
35:         else if w0 ended with 'ed' then
36:             restore 'ed' to w
37:         end if
38:     end if
39:     return syllabified w
40: end function
41: function PYPHEN_PLUS_SYLLABIFY(w)
42:     syls ← []
43:     segs ← wordsegment.segment(w)
44:     for each seg in segs do
45:         syls_ ← pyphen.inserted(seg)
46:         syls_ ← merge_novowel_segments(syls_)
47:         if syls_ has only one syllable and syllapy.count(syls_) > 1 then
48:             syls_ ← naive_syllabify(syls_)
49:         end if
50:         extend syls with syls_
51:     end for
52: end function
```

---

## F    Pseudo-code of BPE+ syllabifier

---

**Algorithm 2** Splitting and Processing BPE Tokens with Vowel Constraints

---

1: **function** SPLIT_ON_NONCONSECUTIVE_VOWELS(token)
2:     parts ← [ ]
3:     current ← [ ]
4:     vowel_positions ← [ ]
5:     **for** each character $ch$ in token **do**
6:         append $ch$ to current
7:         **if** $ch \in$ VOWELS **then**
8:             record position of $ch$ in current
9:         **end if**
10:         **if** two or more vowels are non-consecutive **then**
11:             cut before the last vowel
12:             append left substring to parts
13:             reset current and vowel_positions accordingly
14:         **end if**
15:     **end for**
16:     **if** current is not empty **then**
17:         append current to parts
18:     **end if**
19:     **return** parts
20: **end function**
21: **function** ENFORCE_VOWEL_CONSTRAINT(parts)
22:     remove empty parts
23:     new_parts ← [ ]
24:     buffer ← empty string
25:     **for** each part $p$ in parts **do**
26:         buffer ← buffer + $p$
27:         **if** buffer contains a vowel **then**
28:             **if** next part starts with a vowel **or** "E" **then**
29:                 continue without breaking
30:             **else**
31:                 append buffer to new_parts
32:                 reset buffer
33:             **end if**
34:         **end if**
35:     **end for**
36:     **if** buffer not empty **then**
37:         **if** new_parts not empty **then**
38:             merge buffer into the last part
39:         **else**
40:             append buffer as a new part
41:         **end if**
42:     **end if**
43:     **return** new_parts
44: **end function**
45: **function** BPE_PLUS_SYLLABIFY(tokens)
46:     first_split ← apply split_on_nonconsecutive_vowels to each token
47:     final_parts ← enforce_vowel_constraint(first_split)
48:     **return** final_parts
49: **end function**

---

## G    Spectrogram examples on LibriSpeech

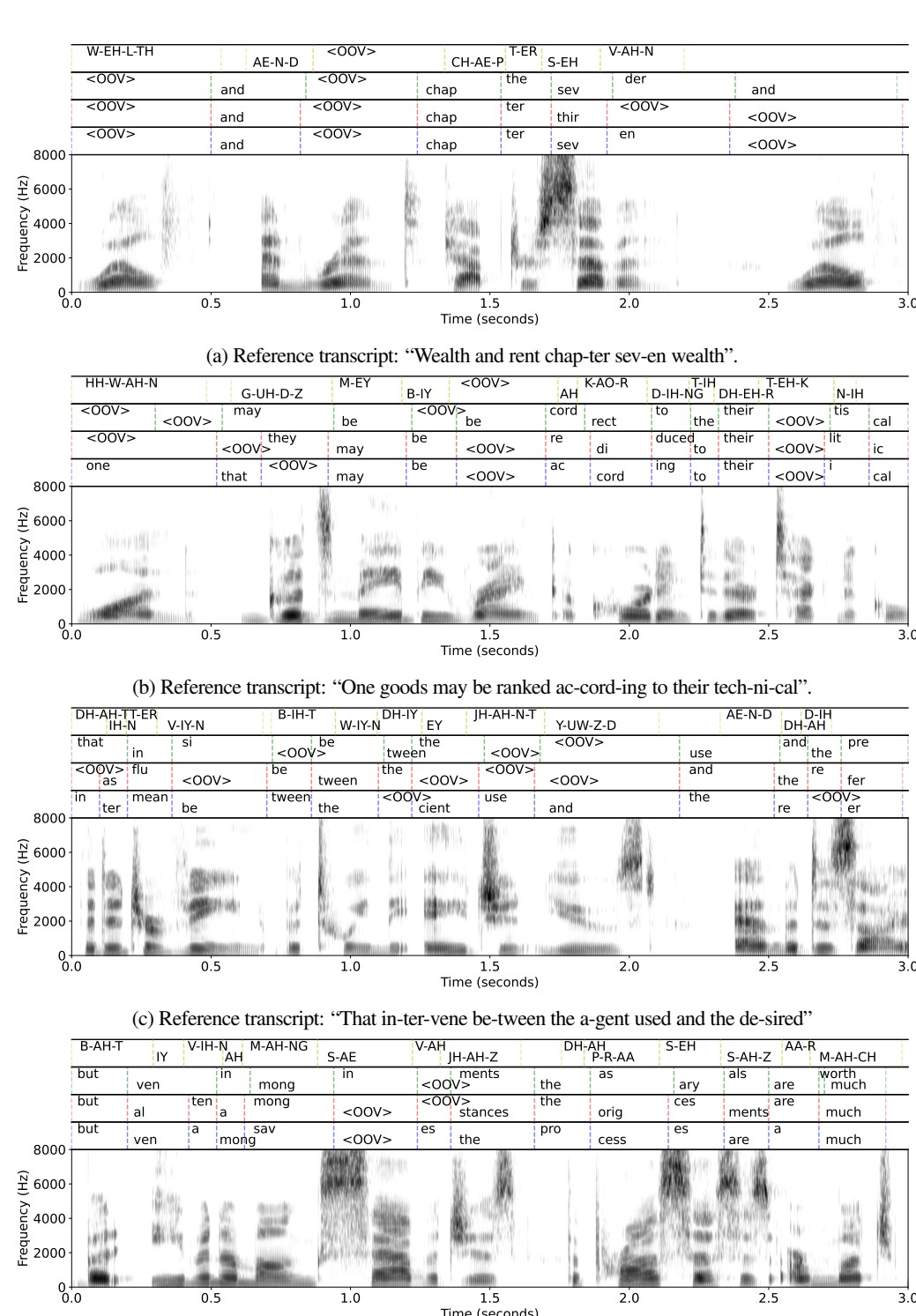

(a) Reference transcript: "Wealth and rent chap-ter sev-en wealth".

(b) Reference transcript: "One goods may be ranked ac-cord-ing to their tech-ni-cal".

(c) Reference transcript: "That in-ter-vene be-tween the a-gent used and the de-sired"

(d) Reference transcript: "But e-ven a-mong sav-ages the pro-cess-es are much"

Figure 4: **Spectrograms of audio examples in our test split of LibriSpeech clean subsets (matched setting) and the predicted speech-text alignment by SylCipher after different training stages.** Audios are truncated to the 3-second mark for better visualization. The alignment bars from top to bottom: Forced alignment, Sylber, Sylber+JE2E, Sylber+JE2E+PUSM.

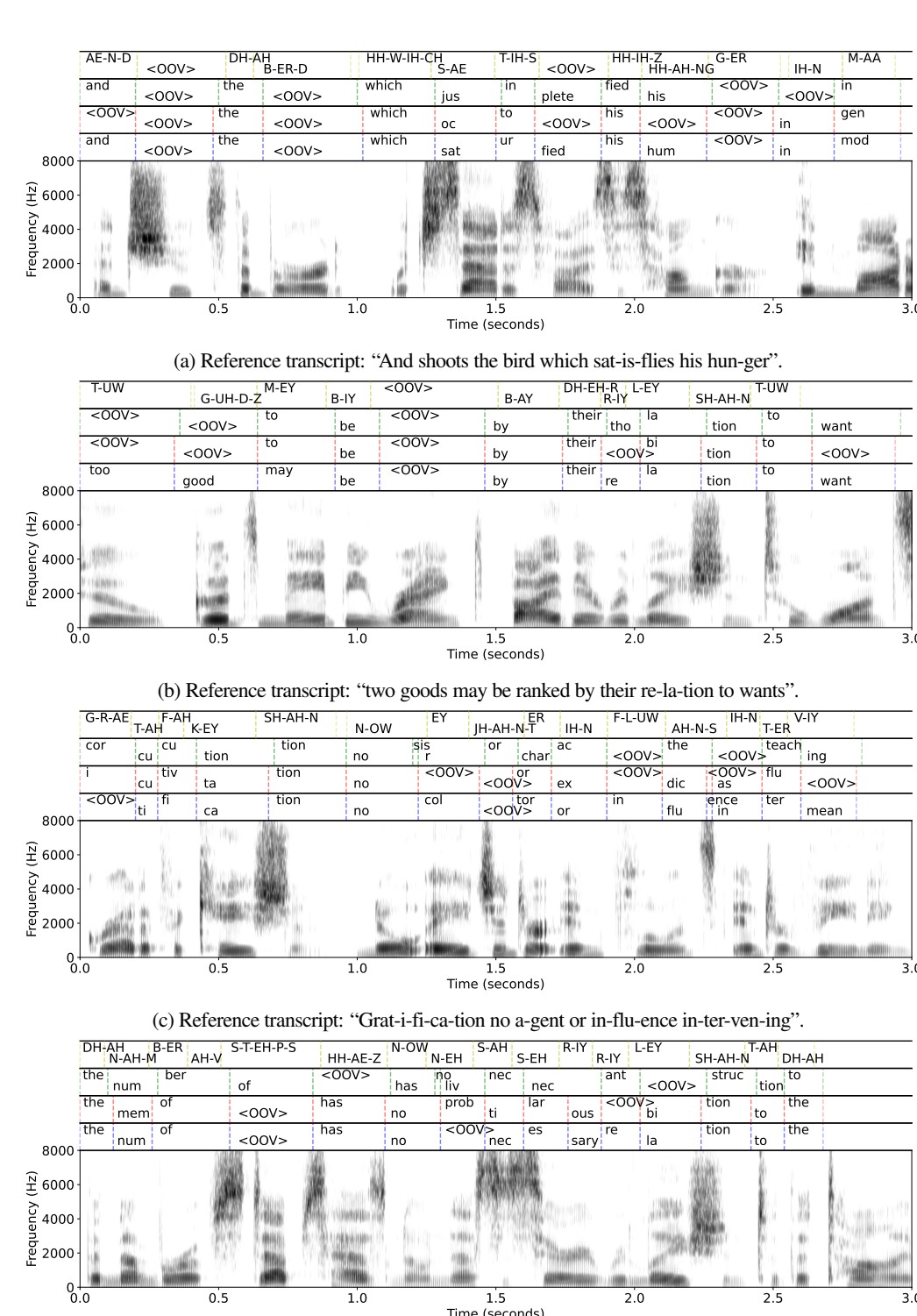

(a) Reference transcript: "And shoots the bird which sat-is-flies his hun-ger".

(b) Reference transcript: "two goods may be ranked by their re-la-tion to wants".

(c) Reference transcript: "Grat-i-fi-ca-tion no a-gent or in-flu-ence in-ter-ven-ing".

(d) Reference transcript: "The num-ber of steps has no nec-es-sary re-la-tion to the".

Figure 5: **Spectrograms of audio examples in our test split of LibriSpeech clean subsets (matched setting) and the predicted speech-text alignment by SylCipher after different training stages.** Audios are truncated to the 3-second mark for better visualization. The alignment bars from top to bottom: Forced alignment, Sylber, Sylber+JE2E, Sylber+JE2E+PUSM.

