# OpenReview forum: "Towards Unsupervised Speech Recognition at the Syllable-Level"
_ICLR.cc/2026/Conference — Submitted to ICLR 2026_

### Official Review · Reviewer_8SKJ · 2025-10-17

**Soundness:** 2
**Presentation:** 3
**Contribution:** 2
**Rating:** 2
**Confidence:** 4

**Summary:**

This paper presents SylCipher, a syllable-based unsupervised ASR (UASR) system that jointly predicts syllable boundaries and embedding tokens from raw speech using a unified self-supervised objective. The authors conduct experiments across domains (LibriSpeech, SpokenCOCO) and languages (English, Mandarin). On Mandarin, SylCipher achieves a 12.2% phone error rate, outperforming previous GAN-based UASR methods.

Overall, this is a paper revisiting the UASR task with a new unit of representation (syllable). While the approach is interesting and the results are solid compared to prior UASR systems, the contribution is mainly the change of processing unit rather than a conceptual or methodological advance. The scope feels somewhat narrow for ICLR and would likely be a better fit for a speech-focused venue such as Interspeech.

Overall, this is a well-executed study with solid experiments, but the novelty and relevance are limited. The work would be more impactful if applied to truly under-resourced languages.

**Strengths:**

* Proposes a new speech unit (syllable) for the UASR problem.
* Achieves competitive results compared to previous UASR approaches.

**Weaknesses:**

* The motivation for UASR is strongest in the context of truly under-resourced languages, but the experiments are limited to English and Mandarin—two well-resourced languages. This makes the work feel a bit outdated, given that by 2025 we expect to see these methods applied to real low-resource settings where they are most needed.

* Limited discussion on whether syllables are suitable units across different language types.

**Questions:**

UASR is not the only approach for building ASR systems for low-resource languages. It would be useful to compare your results with semi-supervised methods, as starting from a small seed (e.g., around one hour of aligned speech–text data) is often realistic even in low-resource or language documentation scenarios.

---

> ### Author Response · Authors · 2025-11-21
>
> We thank the reviewer for the thoughtful comments and for highlighting the strengths of our study. Below we address the concerns regarding novelty, scope, and low-resource relevance, and we provide new empirical results addressing these points.
>
> ---------------------------------------------
>
> **1. Novelty beyond changing the unit**
>
> We respectfully disagree that the contribution is primarily the change of processing unit. The submission proposes **three technical contributions** that go beyond simply swapping phones for syllables.
>
> First, SylCipher is the first to jointly learn boundary probabilities, vector quantization and a shared text–speech masked LM with information-compression constraints a novel explicit distribution matching (PUSM) stage. This is not present in prior UASR work.
> Second, existing soft-poolers (tanh/softmax) are unstable for syllable-level segmentation. Our clamp-based $σ_ε$ yields **6-36% gains across datasets** (Fig. 3b) and avoids overflow/underflow issues (Sec. 4.1). Lastly, we prove (Theorem 1) that MLM-based distribution matching can recover the correct ASR mapping under mild conditions, offering GAN-free, provably stable training (Appendix B). No prior UASR work provides such guarantees.
>
> Thus, the methodological contributions extend well beyond a unit change.
>
> **2. Scope and relevance: applying UASR to truly low-resource settings (New results)**
>
> We appreciate the reviewer’s suggestion to evaluate on **under-resourced languages**. We now include new **Taigi (Southern Min)** experiments, an endangered, under-resourced language, using the SuiSiann corpus.
>
> | **Method** | **Token** | **CER (↓)** |
> |------------|------------------|------------------|
> | wav2vec-U | Phone | 69.0 |
> | SylCipher | Syllable | **37.3** |
>
> The **46% relative improvement** demonstrates that SylCipher works **beyond high-resource languages** in truly low-resource settings, aligning with the reviewer’s stated motivation for UASR. We will include these results in the revision.
>
> **3. Are syllables suitable across language types?**
>
> This is an important question. While syllable structures differ across languages, our experiment provide evidence that syllables are a **flexible cross-lingual unit**. In Mandarin, which has strong syllable-character correspondence; SylCipher achieves **12.2% PER**, outperforming phone-based GAN systems that fail to converge. In English, despite complex syllable structures, syllable modeling yields the best G2P-free CER on LibriSpeech (21.8%) before self-training. In Taigi, which is highly tonal, nasalized, and morphophonemically complex, SylCipher also greatly outperforms phone-level UASR. These results demonstrate that syllable-level modeling generalizes across **three typologically diverse families**. Additionally, we show robustness to **BPE+**, a purely language-agnostic syllable-like tokenizer (Fig. 3a), meaning the success is **not tied to linguistically privileged languages**. We will add these discussions to Section 5 of the paper.
>
> **4. Comparison with semi-supervised learning (New results)**
>
> We fully agree that many low-resource scenarios has **~1 hour of labeled data**. We now include new experiments demonstrating that SylCipher provides **superior pseudo-labels** and integrates seamlessly into semi-supervised pipelines. Our approach based on SylCipher after self-training further finetunes the model with the labeled data. We compare our approach with wav2vec 2.0 and wav2vec-U finetuned in the same manner as ours.
> | **Method** | **Student** | **Labeled Data (h)** | **Matched CER (↓)** |  **Unmatched CER (↓)** |
> |------------|------------------|------------|------------------|------------|
> | wav2vec 2.0 | none | 1 | 22.0 | 22.0 |
> | wav2vec-U | wav2vec 2.0 | 1 | 16.0 | 16.2 |
> | SylCipher (ours) | wav2vec 2.0 | 0 | 17.5 | 33.3 |
> | SylCipher (ours) | wav2vec 2.0 | 1 | **6.8** | **11.1** |
>
> With **only 1 hour** of labeled data, SylCipher reaches **6.8% CER**, significantly outperforming both wav2vec 2.0 and wav2vec-U. This directly answers the reviewer’s question and confirms that UASR systems can be strong foundations for semi-supervised ASR.
>
> **5. On the claim that the work “fits better at Interspeech”**
>
> We respectfully argue that the paper is well aligned with ICLR’s long-standing interest in unsupervised learning, distribution matching, self-supervised speech models, and theoretical insights into representation learning.
> SylCipher contributes to all of these areas and provides a **new theoretical and algorithmic perspective** not tied to a specific task like UASR but offers insights into multimodal self-supervised learning, distribution matching, and compositional representation learning topics that have historically been well received at ICLR.
>
> ---------------------------------------------
>
> We thank the reviewer for helping strengthen the paper. We hope these additions address the reviewer’s concerns and improve the clarity and impact of the work.

---

> > ### Comment · Reviewer_8SKJ · 2025-11-21
> > **acknowledgement of authors' answers**
> >
> > I thank the authors for their response. I particularly appreciate the additional experiment applying their method to a genuinely under-resourced language (Taigi), even though the description and scope of this experiment remain limited. I also acknowledge the inclusion of a comparison to a semi-supervised method. I will raise my score, but it will remain slightly below the acceptance threshold, as I still believe that, to be fully convincing in 2025, such a paper should demonstrate a more extensive application to several truly under-resourced languages.

---

> > > ### Author Response · Authors · 2025-11-28
> > >
> > > Thank you for your thoughtful follow-up. We appreciate your recognition of the Taigi experiment and the semi-supervised comparison, and we fully understand the desire for a broader multilingual evaluation.
> > >
> > > We would like to clarify that the primary contribution of this work is to introduce a **theoretically grounded framework** for G2P-free, syllable-level UASR, rather than to provide an exhaustive empirical survey across many low-resource languages. We believe the conceptual impact of introducing a principled syllable-level training paradigm for G2P-free UASR stands independently of a broad multilingual evaluation, and can serve as a foundation for future work on self-supervised representation learning with speech. Further, the experiments in English, Mandarin, and Taigi demonstrate its generalizability and robustness across **typologically diverse** languages, requiring no G2P or phoneme inventory. The consistent improvements over all G2P-free baselines, including in a genuinely low-resourced setting, indicate that the proposed paradigm generalizes beyond the specific languages evaluated.
> > >
> > > We appreciate the reviewer’s perspective, and we agree that extending the framework to additional low-resource languages is a valuable direction for future work.

---

### Official Review · Reviewer_Y2mg · 2025-10-31

**Soundness:** 3
**Presentation:** 2
**Contribution:** 3
**Rating:** 6
**Confidence:** 2

**Summary:**

This paper tackles the task of unsupervised speech translation, where no paired speech and transcript are included in the training data. The vast majority of UASR systems operate at the phoneme level, which calls for a grapheme-to-phoneme converter. This is not always available for all languages. This paper instead proposes to operate on syllable level.

The overall training paradigm comes in three parts:

1. Use an unsupervised syllable detector to train with an initial boundary
2. Joint E2E training where the soft-polling mechanism is trained along side the rest of the model, hence improving the syllable boundaries
3. Positional unigram and skipgram matching (PUSM) training, which further improves the model with explicit distribution matching between the speech feature and text tokens.

Experiment result shows that the system is able to achieve better performance than all the existing G2P-free approaches, but was unable to push beyond G2P-based method. The method is also shown to be robust in terms of domains and also works for Mandarin.

**Strengths:**

1. Being able to get rid of pronunciation dictionary will significantly help with expanding UASR systems to more low-resource languages.
2. The performance improvement over existing G2P-free approaches is quite remarkable.
3. I liked the fact that the experiments covered both matched and unmatched setups, and examined robustness to domains, scripts, and syllable boundary detection model.

**Weaknesses:**

1. I understand that SylCipher is not entirely language-universal yet because different writing systems exist, but since one of the main benefit of getting rid of pronunciation dictionary was to save work when expanding to new languages, I think it is important to test this on more languages than just English and Chinese.
2. The presentation of the paper could be significantly improved. Here are my suggestions:

* 2a. Re-draw Figure 1 and properly combine the parts that are shared between speech and text inputs.

* 2b. Formula (1) doesn't quite help me understand how I can recover $y^\*$ given only unpaired $X$ and $Y$. You probably want to think about if you can re-write $y^\*$ as an argmax of an objective function that you are optimizing.

* 2c. There are quite few places where aggressive space hacks were used, which I believe you should remove. I would consider removing the theoretical guarantee on Page 4 to make some space for that.

* 2d. I could use a little more intuition apart from Formula 10 as to why PUSM provided more explicit inductive bias for distribution matching.

* 2e. L357: "..., ~~though~~ it's performance is limited by ..."

**Questions:**

1. Why is the evaluation constrained to English? In other words, what's the hurdle to evaluating this also on French? Spanish? Italian? Finnish?
2. L140: I don't follow the last sentence of the paragraph ("As shown in Wang et al. (2023b), ..."). Can you explain?
3. Table 2: I'm confused about why you switched from CER to PER for Mandarin?

---

> ### Author Response · Authors · 2025-11-21
>
> We thank the reviewer for the thoughtful and constructive feedback. We are glad that the reviewer found the contribution meaningful, especially the removal of G2P dependency, the strong G2P-free performance, and the extensive robustness evaluations. Below we address each point in detail.
>
> ---------------------------------------------
>
> **1. Evaluating on additional languages (New results)**
>
> We agree, and we have now conducted **new experiments on Taigi** using the **SuiSiann** corpus. Taigi is a **truly low-resource and morphophonemically complex** language, precisely the type of setting where pronunciation dictionaries are unavailable and UASR is most needed.
>
> | **Method** | **Token** | **CER (↓)** |
> |--------------------|------------------|--------------------|
> | wav2vec-U | Phone | 69.0 |
> | SylCipher | Syllable | **37.3** |
>
> This **46% relative improvement** demonstrates that SylCipher extends effectively beyond English and Mandarin, even those with no standardized orthography. These results will be included in the updated version.
>
> We also highlight that SylCipher’s performance tends to show **larger relative gains on smaller datasets**, supporting its applicability to mid- and low-resource scenarios.
>
> **2. Why not French, Spanish, Italian, Finnish?**
>
> There is no fundamental obstacle. SylCipher is designed to work **without language-specific resources**, and BPE+ (Fig. 3a) already demonstrates robustness with language-agnostic segmentations. We focused our resources on conducting thorough experiments on English, Mandarin, and the newly added Taigi dataset, which cover typologically diverse language families and directly addresses the low-resource motivation of the paper.
>
> **3. Clarification on L140**
>
> The sentence summarizes the key insight from Wang et al. (2023b): **Unsupervised mapping between speech and text is not identifiable in general**, but if syllable boundaries are known (or approximately known), and if the language satisfies mild distributional assumptions, then the mapping **becomes uniquely recoverable**.
>
> We will rewrite this sentence for clarity in the camera-ready version.
>
> **4. Why CER for English vs. PER for Mandarin?**
>
> Mandarin characters correspond to **syllables**, and each syllable decomposes into **initial + final + optional tone**. Therefore, PER offers a tokenization-independent measure of error that properly reflects tonal and phonotactic correctness and is standard in Mandarin ASR evaluation.
>
> **5. Why is evaluation constrained to English?**
>
> It is not – our paper includes **Mandarin**, and we now add **Taigi**, showing robust multilingual generalization without G2P.
>
> **6. Presentation improvements**
>
> We appreciate the reviewer’s detailed presentation suggestions. We will incorporate the following changes:
>
> (1) Redrawing Figure 1 by unifying speech/text shared components and simplifying the diagram to highlight the shared encoder and three training stages more clearly.
>
> (2) Rewrite Eq. (1) to explicitly show that UASR is cast as a distribution-matching problem, where $y^*(x)=\arg\max_y q_{Y|X}(y|x)$ and $q_{Y|X}=\arg\min_q D_{KL}(p_Y||E_{p_X}[q_{Y|X}])$.
>
> (3) Move the theorem statement to the appendix while keeping the intuition in the main text.
>
> (4) We will add an explanation on PUSM that: PUSM explicitly matches positional unigram and local skip-gram statistics, which enforces alignment of short-range text dependencies – something MLM implicitly models but does not explicitly align across modalities. This provides a stronger and more stable inductive bias for speech-text mapping.
>
> (5) Fix any remaining typos.
>
> ---------------------------------------------
>
> We thank the reviewer for acknowledging the importance of the problem and the strengths of our results. We believe these enhancements substantially strengthen the contribution and address the reviewer’s concerns.

---

> ### Comment · Reviewer_Y2mg · 2025-11-27
>
> I thank the authors for the response, especially the low-resource experiments on Taigi. I remain skeptical about the generality of the method due to the limited language coverage.
>
> I also have some reservations about the Chinese experiment. The authors pointed out that:
>
> > Mandarin characters correspond to syllables, and each syllable decomposes into initial + final + optional tone.
>
> That's true, but it's a many-to-one correspondence, and there's no way you can deterministically recover Chinese characters from the pronunciation. An ASR system that outputs Chinese Pinyin is not directly deployable in real-world applications, so I doubt if it's really "standard in Mandarin ASR evaluation". Besides, because of this choice, I also wonder if there is some limitation of the method that make it hard to directly generalize to hieroglyph that wasn't clearly laid out in the paper.
>
> But none of these concerns overshadows the strengths of the paper, so I plan to maintain my score.

---

> > ### Author Response · Authors · 2025-11-28
> >
> > We thank the reviewer for the engaging discussion and for acknowledging the value of our Taigi experiments.
> >
> > Regarding the concern about Mandarin evaluation: We fully agree that Pinyin output is not sufficient for a deployed product due to the homophone issue. We utilized Pinyin/phone error rate to measure the model’s ability to **capture phonetic structures without the confounding factor** of a heavy language model required for character selection.
> >
> > This choice **does not imply a limitation in handling logographic**; rather, it allows us to evaluate the acoustic alignment more directly. We appreciate you highlighting this ambiguity, and we will ensure the camera-ready version explicitly states that **a separate lexical decoding stage would be required** for a user-facing Mandarin application.

---

### Official Review · Reviewer_4Gvh · 2025-11-01

**Soundness:** 3
**Presentation:** 2
**Contribution:** 3
**Rating:** 6
**Confidence:** 4

**Summary:**

This paper introduces SylCipher, which is a novel UASR system that operates at the syllable level rather than the phoneme level, eliminating the need for G2P convertors. The approach uses an iterative training approach and experiments across two English domains and Mandarin show substantial improvements over existing G2P-free approaches. While the approach still requires language-specific syllabification methods to obtain strongest performance and its theory relies on idealized assumptions, the work makes a clear contribution by establishing syllable-level UASR as a viable alternative to phoneme-based systems.

**Strengths:**

Motivation for the study is clearly stated and the manuscript aims to tackle the important problem of making ASR more inclusive of the world’s languages and enabling multimodal learning from non-parallel data. The paper makes a solid contribution by demonstrating that syllable-level modeling can be effective for UASR. The empirical results are strong and the work includes thoughtful ablations on syllabifier choice, pooling mechanisms, and vocabulary size, and evaluates cross-domain robustness. The iterative training procedure, while complex, shows clear improvements at each stage, and the boundary refinement mechanism successfully improves upon the teacher model’s boundaries.

**Weaknesses:**

The paper makes a clear case that UASR is a crucial step toward extending ASR to low-resource languages in the long-tail distribution. However, experiments involving such low-resource languages are missing from the paper, and instead the authors perform experiments on high-resource languages for which well-performing ASR systems exist. Empirical evidence on low-resource languages would substantially strengthen the contribution of this work.

The paper shows empirically that the system works with approximate syllable boundaries, but the theorem has much stronger assumptions and provides no formal analysis of why the theoretical guarantees should hold when its core assumptions are violated. The paper would benefit from such an analysis.

Citation style is inconsistent throughout the text. Narrative and parenthetical forms are sometimes applied in the wrong context. Please standardize. This is a minor issue.

**Questions:**

Could the authors provide results on at least one low-resource language to validate that the method works in the target scenario?

Relatedly, could the authors provide insight into how the amount of training data affects performance of their method?

The performance gains of SylCipher are clear, but the best-performing model requires a 3-stage training procedure. Could the authors provide an analysis of computational costs of training time compared to baselines? That would help assess the practicality of their method.

In Section 4.2, the authors mention using the first transformer layer instead of the last improves performance, citing "over-contextualization" as the cause. Could the authors provide the results for both layer 1 and layer 2 outputs for inference? Does this pattern hold across all three datasets?

---

> ### Author Response · Authors · 2025-11-21
>
> We thank the reviewer for the constructive and detailed feedback. We appreciate the recognition of the importance of our method. Below we address each point raised and provide new experiments that strengthen the contribution.
>
> —---------------------------------------------
>
> **1. Low-resource language experiments (New results)**
>
> We have now conducted **new preliminary experiments** on **Taigi**, a truly low-resource language, using the **SuiSiann** corpus. Taigi has complex morphophonemic processes, making it a realistic stress test for UASR.
>
> | **Method** | **Token** | **CER (↓)** |
> |------------|------------------|------------------|
> | wav2vec-U | Phone | 69.0 |
> | SylCipher | Syllable | **37.3** |
>
> This **46% relative improvement** confirms that SylCipher’s advantages extend beyond high-resource settings and that **syllable-level UASR is effective in exactly the low-resource, real-world scenarios** highlighted in the paper’s motivation. We will integrate these results into the updated version.
>
> **2. Data-efficiency of SylCipher**
>
> We appreciate the reviewer’s question regarding the effect of training data size. Across our three corpora – LibriSpeech (460h), SpokenCOCO (742h), and AISHELL-3 (85h) – we observe that SylCipher continues to improve relative to baselines even as the total amount of unpaired speech decreases. In fact, the largest relative gains occur on AISHELL-3, and our new Taigi experiment (~7h) further confirms that the method remains stable and effective under significantly reduced-resource settings. This suggests that **syllable-level UASR is comparatively data-efficient**, especially compared to GAN-based methods, which we found require substantially larger corpora to stabilize training.
>
> **3. Theory vs. practical assumptions**
>
> We agree that the theoretical assumptions are idealized. In the revised version, we will add a discussion explaining why the guarantees still provide practical value. First, boundary refinement significantly reduces the gap from ideal conditions, as JE2E improves boundary quality over the teacher model by 15% relative F1 (20ms) on LibriSpeech and 16% relative F1 (50ms) on SpokenCOCO. Thus, the model moves itself closer to the idealized regime assumed by the theorem. Further, distribution matching is robust to moderate boundary noise, since the MLM objective and PUSM matching operate on **token distributions**, not exact segment lengths. Therefore, misaligned boundaries lead to **local perturbations**, but distributional signals remain stable as long as the number of segments approximates the true number. We will add these insights to Section 4 and Appendix B.
>
> **4. Computational cost of 3-stage training**
>
> Yes, We will add the following computational comparison in the revision:
>
> | **Model** | **GPUs** | **Hours** |
> |------------|------------------|------------------|
> | wav2vec-U | 8$\times$V100 | ~48 |
> | SylCipher (fixed) | 8$\times$V100 | ~24 |
> | SylCipher (JE2E) | 8$\times$V100 | ~12 |
> | SylCipher (PUSM) | 8$\times$V100 | ~12 |
>
> Despite the iterative design, **SylCipher is comparable to wav2vec-U** in terms of training time for two main reasons. First, it avoids adversarial training and grid search with the regularization weights. Second, both the MLM phases and the PUSM stage converge quickly and stably. We will include this table in the revision.
>
> **5. First-layer vs. last-layer inference results**
>
> Thank you for pointing this out. After re-checking our experiments, we found that our claim in Section 4.2 was based on an earlier set of runs and does not fully hold in the final version of the experiments.
>
> | **Layer** | **LibriSpeech (CER)** | **SpokenCOCO** | **AISHELL-3** |
> |------------|------------------|------------|------------------|
> | 1 | **21.4** | 27.1 | **26.9** |
> | 2 | 21.8 | **26.8** | 31.9 |
>
> As the reviewer observes, **Layer 1 performs best for LibriSpeech and AISHELL-3**, while **Layer 2 performs best for SpokenCOCO**. We will correct this statement in the revised version.
>
> Rather than a universal rule, the more accurate takeaway is that: Early-layer representations are helpful when syllabic alignment requires finer-grained local information, while later-layer representations can be beneficial under cross-domain mismatch, where increased contextualization may compensate for domain differences (e.g., SpokenCOCO).
> We will revise the paper to reflect this balanced interpretation.
>
> Importantly, this correction does **not** affect the main findings of the paper, since SylCipher’s performance gains remain unchanged, and all baselines and ablations remain valid.
>
> **6. Citation style**
>
> We thank the reviewer for noting this. We will standardize all citations to consistent parenthetical style per ICLR guidelines.
>
> ---------------------------------------------
> We thank the reviewer again for the positive assessment of the soundness and contribution. We hope these additions address all concerns and further strengthen the contribution of the paper.

---

### Official Review · Reviewer_dnBS · 2025-11-03

**Soundness:** 2
**Presentation:** 3
**Contribution:** 3
**Rating:** 4
**Confidence:** 4

**Summary:**

The paper introduces SylCipher, a novel framework for unsupervised automatic speech recognition (UASR) at the syllable level. Unlike prior UASR systems that operate at the phoneme or word level (often requiring grapheme-to-phoneme converters or suffering from instability), SylCipher leverages masked language modeling and recent advances in unsupervised syllable boundary detection to align unpaired speech and text. The approach is resource-efficient (no G2P required), theoretically grounded, and empirically validated across English (LibriSpeech, SpokenCOCO) and Mandarin (AISHELL-3). SylCipher achieves up to a 40% relative reduction in character error rate (CER) over previous G2P-free UASR methods and demonstrates robustness to domain and language shifts.

**Strengths:**

The strengths of the paper are:
1. Originality:
   - First to propose syllable-level UASR, bridging the gap between phoneme- and word-level approaches.
   - Avoids reliance on G2P or pronunciation dictionaries, making it applicable to truly low-resource languages.
   - Theoretical analysis provides guarantees for distribution matching and zero-error UASR under certain conditions.

2. Quality:
   - Extensive experiments on multiple datasets (LibriSpeech, SpokenCOCO, AISHELL-3) and languages (English, Mandarin).
   - Careful ablation studies on segmentation, vocabulary size, and pooling mechanisms.
   - Robustness to domain and language shifts is empirically demonstrated.

3. Clarity:
   - Clear motivation for syllable-level modeling and its advantages over phoneme/word-level approaches.
   - Well-organized presentation of architecture, training objectives, and results.
   - Visualizations and tables directly support the claims.

4. Significance:
   - Enables unsupervised ASR in languages lacking G2P resources.
   - Advances the state of the art in UASR, with potential impact on speech technology for low-resource and endangered languages.

**Weaknesses:**

The weaknesses of the paper
1. The method is not yet language-universal; languages with non-syllabic scripts (e.g., Hebrew, Arabic) may pose challenges for syllabification.

2. While the method is robust to segmentation noise, performance still depends on the quality of unsupervised syllable boundary detection.
The approach to syllabification (e.g., Pyphen+, BPE+) may require further tuning for new languages.

3. The iterative training procedure (fixed boundary, JE2E, PUSM) could be further simplified into a fully end-to-end approach.

**Questions:**

The work is with good quality in general. I have several questions:

1. In natural language processing nowadays, languages are covered into one tokenizer. For speech, phone-level modeling can somehow cover most of the languages by sharing one pronunciation vocab, e.g.. How about the syllable-level? Is it possible to train a multi-lingual UASR system by leveraging a more universal/ language agnostic speech tokenizer? It is interesting to see the method effective on more than single language individually.

2. Based on the results in table 1, REBORN still holds the best performance in unmatched case. Does it mean phone-based method still has the advantage?

3. How to perform down-stream ASR finetuning given limited paired data? This would bring the system into a more practical usage. Any experiment to show the effectiveness of the ASR finetuning?

---

> ### Author Response · Authors · 2025-11-21
>
> Thank you for the constructive feedback and for acknowledging the strengths of our paper. We address the main concerns raised:
>
> ---------------------------------------------
>
> **1. Can a more universal/multilingual tokenizer enable a multilingual UASR?**
>
> Yes. While full language-universality remains open, our results already demonstrate that syllable-level modeling generalizes across **three typologically diverse families** – Germanic (English), Sinitic (Mandarin), and now **Southern Min / Taigi**, a low-resource tonal language with code-switching.
>
> We conducted preliminary experiments on the **SuiSiann Taigi dataset** (5% code-switching with English). Despite the language’s complex syllable structure and mixed-script representation, **SylCipher significantly outperforms wav2vec-U**:
>
> | **Method** | **Token** | **CER (↓)** |
> |--------------|------------|-------------|
> | wav2vec-U | Phone | 69.0 |
> | SylCipher | Syllable | **37.3** |
>
> This 46% relative improvement shows that **syllable-level modeling is more language-agnostic and stable than phoneme-based modeling in the absence of G2P** for mixed-script or tonal low-resource languages.
>
> Further, our architecture (shared encoder + MLM + PUSM) aligns well with multilingual MLM approaches. BPE+, a completely language-agnostic tokenizer, yields competitive results (Fig. 3a), suggesting that a **universal syllable-like tokenizer** is feasible.
>
> SylCipher’s strong results on English, Mandarin and Taigi confirm that syllables provide a cross-lingual alignment unit, unlike phonemes, which often misalign due to language-specific inventories and scripts.
>
> We agree with the reviewer that extending SylCipher to a multilingual UASR setting is a natural next step.
>
> **2. Does REBORN outperform SylCipher in the unmatched setting?**
>
> Only in the **G2P-based** setting.
> The strong REBORN result cited is from the G2P-trained phoneme model (REBORN*).
> In the **G2P-free** scenario (the main focus of UASR), REBORN degrades substantially:
>
> | **Method** | **CER (↓)** |
> |--------------|------------|
> | REBORN (no G2P) | 76.6 |
> | SylCipher | **35.9** |
>
> Thus, phoneme-based UASR is competitive only **when pronunciation resources are available**. Under the intended G2P-free UASR regime, **syllable-level modeling is far more robust**, especially in low-resource and cross-lingual cases (Taigi, Mandarin).
>
> **3. How can the model be finetuned with limited labeled data?**
>
> Thank you for the suggestion. We now include **new semi-supervised finetuning experiments**, demonstrating that SylCipher provides superior pseudo-labels for downstream ASR. The experiment is conducted on LibriSpeech with the same split as in the paper. Our approach based on SylCipher after self-training further finetunes the model with the labeled data. We compare our approach with wav2vec 2.0 and wav2vec-U finetuned in the same manner as ours.
>
> | **Method** | **Student** | **Labeled Data (h)** | **Matched CER (↓)** |  **Unmatched CER (↓)** |
> |------------|------------------|------------|------------------|------------|
> | wav2vec 2.0 | none | 1 | 22.0 | 22.0 |
> | wav2vec-U | wav2vec 2.0 | 1 | 16.0 | 16.2 |
> | SylCipher (ours) | wav2vec 2.0 | 0 | 17.5 | 33.3 |
> | SylCipher (ours) | wav2vec 2.0 | 1 | **6.8** | **11.1** |
>
> With just 1 hour of paired data, SylCipher achieves **6.8% CER**, outperforming both wav2vec 2.0 wav2vec-U by a large margin. Even without labeled data, SylCipher already produces high-quality pseudo-labels. This confirms SylCipher’s strong compatibility with **practical low-resource ASR**.
>
> **4. Dependence on syllabification and segmentation quality**
>
> Thank you for pointing this out. Although syllable boundary quality impacts performance, our JE2E stage consistently improves boundaries **beyond** the teacher by 15% relative F1 (20ms) on LibriSpeech and 16% relative F1 (50ms) on SpokenCOCO, as shown in Table 3. This demonstrates that **unpaired text distributions guide better segmentation**, reducing dependence on initial syllabifiers.
>
> **5. Complexity of iterative training**
>
> We acknowledge the reviewer’s comment. The staged pipeline is motivated by stability (PUSM requires full-dataset batches), but we are exploring joint end-to-end training and curriculum-based versions that simplify training while keeping theoretical guarantees.
>
> ---------------------------------------------
>
> Thank you again for the thoughtful comments and hope our response helps to address your concern and makes our paper better suited for ICLR. Happy to discuss further and consider further revisions.

---

### Meta-Review · Area_Chair_QpDM · 2026-01-03

**Summary:**

There exist a few key concerns that informed my suggested decision for this paper: (i) The scope feels somewhat narrow for ICLR—the authors are simply replicating their already proposed UASR idea to new units, namely syllables. The results are better, but the theory is largely the same. (ii) The authors mainly focus on resource-rich languages, for which UASR is practically useless. This comment is shared by all reviewers. The authors provide new results on a resource-poor language (Taigi), but more extensive application to several truly under-resourced languages is needed. (iii) There is no formal analysis of why the theoretical guarantees should hold when the core assumptions are violated.

4Gvh points out that the submission does not follow the ICLR format and, according to the guidelines, it should be desk-rejected. Indeed, the “Anonymous authors” and “Paper under double-blind review” lines under the title are missing.

**Reviewer Concerns:**

The authors have addressed some of the concerns raised at the end of the review process, for example: (i) REBORN’s superiority over the proposed method, and (ii) the use of paired data for fine-tuning. However, key critical concerns remain outstanding: (i) narrow scope and limited novelty with respect to what has already been proposed as UASR, (ii) limited experimental assessment of the idea on several truly under-resourced languages, and (iii) a not fully convincing response regarding the theoretical guarantees.

**Reviewer Scores:**

dnBS might have kept the score unchanged.

4Gvh might have slightly increased their score because the authors have shown results on one under-resourced language, as asked by the reviewer. However, this reviewer also suggested a desk rejection due to formatting issues.

Y2mg might have either kept their score or increased it by 1.0—not sure; their comments are not very technical. Their comments indeed suggest a  limited familiarity with the problem. Accordingly, their opinion should be weighted with some caution when formulating the final decision.

8SKJ would have increased it as they stated in their rebuttal. However, the increase in their score clashes  with their final statements: (i) "the description and scope of this experiment remain limited," referring to the Taigi experiment provided in the rebuttal; and (ii) "I still believe that, to be fully convincing in 2025, such a paper should demonstrate a more extensive application to several truly under-resourced languages."Accordingly, their opinion should be weighted with some caution when formulating the final decision.

---

### Decision · Program_Chairs · 2026-01-26

Reject